# Pathological Analysis of Encased Resected Recurrent Nerves in Locally Invasive Thyroid Cancer

**DOI:** 10.3390/cancers14122961

**Published:** 2022-06-15

**Authors:** Alexandre Dahan, Abir Al Ghuzlan, Randa Chehab, Joanne Guerlain, Ingrid Breuskin, Camilo Garcia, Livia Lamartina, Julien Hadoux, Eric Baudin, Dana M. Hartl

**Affiliations:** 1Thyroid Surgery Unit, Department of Anesthesia, Surgery, and Interventional Radiology, Institute Gustave Roussy, 94805 Villejuif, France; alexandredahan@gmail.com (A.D.); joanne.guerlain@gustaveroussy.fr (J.G.); ingrid.breuskin@gustaveroussy.fr (I.B.); 2Department of Biology and Pathology, Institute Gustave Roussy, 94805 Villejuif, France; abir.alghuzlan@gustaveroussy.fr (A.A.G.); randa.chehab@curie.fr (R.C.); 3Department of Nuclear Medicine and Endocrine Oncology, Institute Gustave Roussy, 94805 Villejuif, France; camilo.garcia@gustaveroussy.fr (C.G.); livia.lamartina@gustaveroussy.fr (L.L.); julien.hadoux@gustaveroussy.fr (J.H.); eric.baudin@gustaveroussy.fr (E.B.)

**Keywords:** thyroid cancer, recurrent nerve, vocal fold paralysis, aggressive subtypes, high-risk

## Abstract

**Simple Summary:**

Thyroid cancer encasing the recurrent nerve is rare, and the decision to resect or preserve the nerve is multifactorial. The aim of this retrospective study was to evaluate the rate of actual invasion of the nerve beyond the nerve sheath in cancers encasing the nerve. Fifty-two patients were included: 7 cases of medullary thyroid carcinoma, 8 papillary thyroid carcinomas in children and 37 follicular derived cancers in adults. Tumor-related vocal fold paralysis was present in 30% of cases. The nerve was invaded in 82% of follicular cell-derived tumors, 88% of pediatric cases, and 100% of medullary carcinomas. Only agressive histology was a risk factor for nerve invasion. Vocal fold paralysis was not predictive. To our knowledge, this is one of the largest series with pathologic analysis of resected recurrent nerves, showing a high rate of nerve invasion in these rare cases of cancer encasing the reucrrent nerve.

**Abstract:**

Objective: Thyroid cancer encasing the recurrent nerve is rare, and the decision to resect or preserve the nerve is multifactorial. The objective of this study was to histopathologically analyze resected encased nerves to assess the rate of nerve invasion and risk factors. Materials and Methods: A retrospective study was carried out on consecutive patients with resection of the recurrent nerve for primary or recurrent follicular cell-derived or medullary thyroid carcinoma from 2005 to 2020. Demographics, pathology, locoregional invasion, metastases, recurrences and survival were analyzed. Slides were reviewed blindly by two specialized pathologists (AAG, RC) for diagnosis of invasion deep to the epineurium. Results: Fifty-two patients were included: 25 females; average age, 55 (range 8–87). In total, 87% percent (45/52) were follicular cell-derived with 17/45 (37.8%) aggressive variants; 13% (7/52) were medullary carcinoma. Preoperative vocal fold (VF) paralysis was present in 16/52 (30.7%). Pathologically, the nerve was invaded in 44/52 cases (85%): 82% of follicular cell-derived tumors (37/45), 88% of pediatric cases, and 100% of medullary carcinomas (7/7). Nerve invasion was observed in 11/16 (69%) with preoperative VF paralysis and 33/36 (92%) with normal VF function. Only aggressive histology was correlated with nerve invasion in follicular cell-derived tumors (*p* = 0.019). Conclusions: The encased nerves were pathologically invaded in 82% of follicular cell-derived tumors and in 100% of medullary carcinomas. Nerve invasion was statistically correlated with aggressive histopathological subtypes and was observed in the absence of VF paralysis in 92% of cases.

## 1. Introduction

Thyroid cancer encasing the recurrent nerve is a rare entity and the decision to resect or to preserve the nerve depends on many factors. Current guidelines propose decision algorithms and criteria to consider when performing a risk–benefit analysis between nerve resection and nerve preservation in locally invasive thyroid cancer [1,2,3]. When the nerve is encased, macroscopically complete tumor resection (R0/R1) is generally very difficult without nerve sacrifice and the consequences of leaving macroscopic residual disease, even if small, must be weighed against the benefits of complete resection or the possibility of effective adjuvant therapy.

The objective of this study was to histopathologically analyze encased recurrent nerves in cases where resection had been chosen, in order to assess the rate of true microscopic invasion within the epineurial nerve sheath itself and to evaluate risk factors for deep nerve invasion.

## 2. Materials and Methods

A retrospective study of patients having undergone resection of the recurrent nerve for primary or recurrent follicular cell-derived or medullary thyroid carcinoma from 2005 to 2020 was undertaken. The local institutional review board approved the study and individual patient consent was waived. Patient demographics, pathology, invasion of the visceral axis, distant metastases, local recurrences and survival were analyzed. Pathology was retrospectively reviewed blindly by two specialized pathologists (AAG, RC) for diagnosis of nerve invasion. Criteria for diagnosis of nerve invasion was the presence of tumor cells or stroma in contact with nerve fibers, that is, inside the inner epineurium. (Figure 1). Risk factors for nerve invasion (age, gender, pathology, reoperation, preoperative recurrent nerve paralysis and resection of the visceral axis) were analyzed. A letter containing the Voice Handicap Index (VHI 10) and the European Organization for Research and Treatment of Cancer Quality of Life Questionnaire–Head and Neck 35 (EORTC-QLQ–HN35) questionnaires was mailed to patients using their last known address, with a postage-paid envelope for returning the response. Descriptive statistics and non-parametric statistical tests were performed using the SPSS software (IBM^®^ version 5.0).

## 3. Results

A total of 52 patients were included, 25 females and 27 males, with an average age of 55 years (range 8–87). Eight patients were under 18 years old. 

Follicular cell-derived tumors represented the majority (45/52 or 87%), whereas medullary thyroid carcinoma represented seven cases (13%). Of the 45 follicular cell-derived tumors, 17 (37.7%) were aggressive variants of papillary thyroid carcinoma (13 tall cell, 2 columnar cell and 2 diffuse sclerosing), 6 were Hürthle cell carcinomas, 5 were poorly differentiated carcinoma and the others were classic papillary carcinoma. Sixteen patients (35%) with follicular cell-derived tumors had known radioactive iodine-refractory tumors. The eight pediatric cases were follicular cell-derived tumors (classic papillary *n* = 6, diffuse scerlosing variant *n* = 2). 

Distant metastases were present at the time of nerve resection in 11 patients (21%), 5 of whom were children with lung metastases. The distant metastases in adults were judged to be low-volume lung metastases. All tumors were at a high risk of recurrence according to the 2015 ATA guidelines [4] and 25 patients (48%) were stage IV according to the 8th edition TNM classification [5]. All of the resections were preoperatively approved by our local tumor board, after discussing risks, benefits and alternative therapies. 

Twenty patients were reoperative cases (38%), whereas 32 were primary resections (62%). In children, 7/8 were primary resections, the nerve being encased by the tumor, the lymph nodes or both. There were no cases in which the vocal fold had been paralyzed during a previous operation; preoperatively, the vocal folds were paralyzed due to the primary tumor or the recurrent/persistent disease.

Preoperative vocal fold (VF) paralysis was present in 16/52 cases (30%). No patient had contralateral vocal fold paralysis preoperatively or postoperatively. Concomitant partial resection of the larynx, trachea and/or esophagus was performed in nine cases, but no total laryngectomies were performed. 

Nerve sacrifice was performed in an attempt to optimize locoregional tumor resection with R0/R1 margins [5]. For follicular cell-derived tumors in adults, tumor resection was considered R0 in 49% of cases, minimal R1 in 39% of cases and R2 in four cases (8%). Residual disease in patients with R2 resection was left on the tracheal wall, due to patient comorbidities precluding tracheal resection, and adjuvant external beam radiation therapy was administered to these four patients. A total of 18 patients—all adults, 15 with known radioactive iodine refractory tumors and 3 with medullary carcinoma, had postoperative external beam radiation therapy. All of the pediatric cases and 61% of the adult differentiated thyroid cancer cases received adjuvant radioactive iodine.

On retrospective pathological analysis, the nerve was microscopically invaded in 44/52 cases (85%) (Table 1).

Preoperative vocal fold paralysis was not statistically related to nerve invasion (Fisher’s exact test, *p* = 0.09). The nerve was invaded in 92% of patients without preoperative vocal fold paralysis, and in 69% of patients with preoperative vocal fold paralysis. In the 44 patients with nerve invasion, 33 (75%) had normal preoperative vocal fold mobility.

Concomitant resection of the visceral axis was not statistically related to nerve invasion; in the nine patients with associated tracheal or esophageal resection, seven had deep epineurial invasion (Fisher’s exact test, *p* = 0.61). Age, gender and primary versus reoperation were not statistically related to nerve invasion.

Aggressive tumor histology was the only factor found to be related to nerve invasion for follicular cell-derived carcinomas (*p* = 0.019 Fisher’s exact test). Classic papillary carcinoma invaded the nerve in 14/21 cases (Figure 1 and Figure 2), whereas aggressive variants invaded nerves in 22/23 cases (Figure 3). Evaluating only the adult population, aggressive tumor histology was still associated with nerve invasion (*p* = 0.008). 

Follow-up averaged 58.9 months, with a median follow-up of 54 months (range: 1 to 147 months). The Kaplan–Meier calculated overall survival rate for follicular cell-derived tumors in adults was 97% at 1 year and 86% at 5 years. For these tumors, 44% of patients achieved an excellent response according to the 2015 American Thyroid Association Guidelines, 22% a biochemically incomplete response and 33% a structurally incomplete response [4]. The group of medullary carcinomas was too small to calculate a meaningful actuarial survival rate, and there were no deaths in the pediatric group.

The rate of local recurrence in the region of the resected nerve was 0% at 5 years for patients with follicular cell-derived thyroid cancer and R0/R1 resection. Two of the four patients with R2 resection had local progression at the site of residual disease (trachea) despite adjuvant external beam radiation therapy. For medullary carcinoma, the overall rate of local recurrence at the level of the resected nerve was 14% (1/7). In the pediatric group, follow-up ranged from 18 to 140 months (average 71 months). One local recurrence and one case of recurrent lung metastases occurred, treated with repeated doses of radioactive iodine. The other six of the eight pediatric patients achieved excellent response or biochemically incomplete response, without persistent structural disease at last follow-up.

From a functional standpoint, four adult patients and two pediatric patients required tracheostomy for concurrent laryngo–tracheal resections, but with only one adult patient maintaining a permanent tracheostomy. The two pediatric patients were decannulated after endoscopic laser-assisted medial arytenoidectomy. Intraoperative nerve reconstruction with ansa hypoglossi was performed in only two patients due to the distal resection of the nerve (close to the larynx) in most cases, but also due to the fact that this practice was not routine for the early patients in our study. Secondary vocal fold medialization was performed in five adults—three autologous fat injections and two Isshiki type I thyroplasties [6]—and one child (autologous fat injection). Fifteen adult patients responded to a mailed questionnaire sent an average of 2 years postoperatively for subjective evaluation. The average voice handicap index (VHI 10) [7] for these 15 patients was 19.7, with a median of 17 (range 4–34). The average score on the European Organization for Research and Treatment of Cancer Quality of Life Questionnaire–Head and Neck 35 (EORTC-QLQ–HN35) [8] questionnaire was 60, with a median of 52 (range 35–108).

## 4. Discussion

Locally invasive thyroid cancer is rare, with reported incidences ranging from 4–10% [9,10,11]. Recurrent nerve invasion is the second most frequent type of local invasion after strap muscle invasion [12]. Often, this invasion does not only involve the nerve but also the trachea, larynx and/or esophagus more or less superficially, requiring shaving or segmental resections. Disease-specific survival has been shown to be similar and not significantly different between R0 resections and R1 resections for locally invasive thyroid carcinoma [13,14,15,16,17,18]. However, in a recent study of 72 patients with intraoperative findings of recurrent nerve involvement, a positive resection margin was correlated with recurrence, [19] and completeness of resection is a prognostic factor when using the MACIS risk stratification system for surgical decision making [20]. Nerve involvement has been shown to impact recurrence-free survival, but to a lesser extent than tracheal or esophageal invasion. However, recurrent nerve invasion alone has not been shown to be a factor for disease-specific survival [11,14,17,18,21]. Current guidelines and consensus statements emphasize nerve preservation in order to maintain quality of life, due to the relatively favorable prognosis of differentiated thyroid cancer and outline the multiple factors involved when deciding to preserve or resect a nerve for oncological purposes. (Table 2) [1,2,3,22]. Tumors encasing the recurrent nerve are ATA high-risk lesions, with reported rates of structural recurrence of up to 40–50% [4]. From a survival standpoint, T4a tumors in patients over 55 years of age are stage III with a survival rate of 60–70%, and may warrant more aggressive surgical resection [23]. 

Our study included 11 adult patients with low-volume distant metastases for whom the decision to resect the nerve was based on optimizing local control to avoid local complications of systemic therapies [24]. For the other patients, we favored complete resection to optimize chances of attaining an excellent response. Furthermore, the location of the tumors at the distal end of the recurrent nerve led us to favor complete resection in order to avoid potential tumor progression into the larynx or inferior constrictor muscles.

Aggressive variants and medullary thyroid carcinoma invaded the nerve more often than classic papillary thyroid carcinoma. Classic papillary thyroid carcinoma, particularly in young patients, is also classically responsive to radio-iodine, with high rates of overall survival (25). In these cases, preservation of the nerve with residual tumor and adjuvant radio-iodine (RAI) would thus seem appropriate. Leaving residual macroscopic disease, however, may lead to repeated administrations of RAI with high cumulative doses and risks of short- and long-term toxicity [25,26]. Furthermore, many aggressive variants are RAI-refractory, but for primary resections, the exact pathological subtype is not always known beforehand. Performing preoperative ^18^Fluorodeoxyglucose positron emission tomography with computed tomography (^18^FDG-PET CT) in locally invasive tumors may provide information on tumor aggressiveness and responsiveness to RAI [27,28]. Even for RAI-sensitive tumors, a large remnant may not be completely cured with RAI, its efficacy being at a maximum for small-volume lesions [29]. 

Nerves are composed of axons surrounded by endoneurium, grouped into fascicles by perineurial tissue, surrounded by the inner epineurial space and then enclosed by the outer epineurium [30]. “Perineural invasion” of tumors (not to be confused with the perineurium) was broadly defined in 1985 by Batsakis as tumor cell invasion in, around and through the nerve [31,32]. In a review of the literature published in 2009, Liebeg et al. suggested that “finding tumor cells within any of the 3 layers of the nerve sheath (epineurium, perineurium or endoneurium) or tumor foci outside of the nerve with involvement of ≥33% of the nerve’s circumference” were features defining perineural invasion in malignant tumors [32]. The 2018 International Neuromonitoring study group, guidelines defined neural invasion as involving >33% of the circumference of the nerve [1]. All of our patients met the International Neuromonitoring definition of neural invasion, all having encased nerves with over 33% of the circumference of the nerve surrounded by tumor.

To our knowledge, this is the largest histopathologic study of resected encased recurrent laryngeal nerves. Our high rate of microscopic invasion (85%) may be due to a selection bias of only encased nerves, reoperative cases (40%) and aggressive variants (37%). 

Kihara et al. histologically examined the normal part of three resected recurrent nerves and showed that 78–82% of what the surgeon sees as nerve is actually connective tissue surrounding the nerve: the actual axons only comprise 18–22% of the cross-sectional area of the nerve [33]. This finding justifies partial layer resection as described by Nishida et al. when possible, preserving all or part of the nerve axons [34]. This type of nerve preservation, when possible, has been shown to provide better vocal results than nerve grafting or phonosurgery [35,36].

In our study, 33/36 patients without preoperative VF paralysis had nerve invasion on histologic examination. It is well known that axons can continue to function despite tumor infiltration. On the contrary, however, five patients with preoperative vocal fold paralysis did not have nerve invasion. This again may be due to a lack of complete pathological analysis of the entire nerve, due to the retrospective nature of our study, or these nerves may have been paralyzed by compression, devascularization or inflammation, which we did not pathologically analyze.

With results supporting the results of our study, Chen et al. compared a group of 105 patients with nerve invasion found visually and/or histopathologically with a control cohort of 3131 patients without nerve involvement [37]. The nerves were managed either with partial layer resection or resection and nerve reconstruction. Preoperative vocal fold paralysis was a predictive factor for nerve involvement on multivariate analysis, as compared to the control group, but nonetheless, 73% of the patients in the invaded nerve group did not have preoperative vocal fold paralysis. 

Kamani D. et al. studied 12 patients with preoperative vocal fold dysfunction and histological nerve invasion; intraoperatively, a stimulated electrophysiologic nerve response was observed in 33% of these cases [38]. In the same study, in 22 nerves seen to be grossly invaded during surgery, 45% of patients had normal laryngeal mobility preoperatively and in the remaining 55% with abnormal focal fold function, 33% still had a stimulated electrophysiologic response intraoperatively. In the recent study by Kesby et al., of 154 patients with medullary thyroid carcinoma, 11 patients were found to have recurrent nerve invasion but 5 of these patients (45%) had normal preoperative vocal fold mobility [39].

Our study has many drawbacks due to its retrospective nature. Nerve reconstruction was performed in only a few cases, anastomosing the ansa hypoglossi to the recurrent nerve stump as described by Crumley in 1986 [40] and Miayuchi in 1998 [41], but long-term voice or quality of life data were missing for the majority of our cases, making any kind of comparison impossible. The VHI and EORTC data were only available for 15 patients who responded to the questionnaires. A VHI 10 score above 11 is considered abnormal [42]. The VHI 10 scores observed (19.7 on average, with a median of 17) are similar to those reported for transoral laser-assisted vocal fold resection for laryngeal cancer [43]. The EORTC scores were comparable to average scores observed in patients treated for head and neck squamous cell carcinoma [8]. Today, we attempt ansa hypoglossi nerve anastomosis in all cases in which a small nerve stump can be preserved at the laryngeal entry point. We do not perform intralaryngeal anastomosis when the tumor involves the distal end of the recurrent nerve, but this technique seems to improve functional results when feasible [36]. We did not record quantitative measurements during surgery using intraoperative nerve monitoring. Stimulated amplitudes and latencies in patients with encased nerves but still functioning vocal folds may be informative in predicting nerve invasion versus intact nerve envelopes. Finally, the oncologic results only reflect a very small heterogeneous cohort of rare and highly-selected tumors.

The retrospective nature of this study precluded any more detailed analysis of the invaded nerves, such as length of invasion, ratio of tumors stroma or macrophages versus tumor cells within the nerve sheath, or other aspects that may have allowed us to distinguish distinct groups according to the aspect of nerve invasion. We did not look at imaging to determine the sensitivity or specificity of different imaging techniques to predict nerve invasion. This would require a larger cohort and more patients without nerve encasement for comparison. 

Finally, we were not able to draw precise conclusions as to exactly which factors were predominant in deciding to resect the recurrent nerve, due the heterogeneity and retrospective nature of our study. Each case was discussed preoperatively within our local tumor board and nerve resection (if necessary) was validated. When complete resection (R0/R1) was deemed to be possible by the tumor board (with surgeons attending), nerve resection was validated if necessary to achieve this goal. When the tumor was known to be iodine-refractory (for differentiated thyroid cancer), our tumor board also generally validated nerve resection. Most of these patients, however, were treated before the widespread use of BRAF or RET inhibitors or before neo-adjuvant treatment protocols were employed. Some of these cases today may have been amenable to neo-adjuvant treatment kinase inhibitors, and their nerves may possibly have been spared.

## 5. Conclusions

To our knowledge, this is the largest histopathological evaluation of encased resected recurrent nerves for thyroid cancer. The inner epineurium or beyond was microscopically invaded by tumor cells in 82% of follicular cell-derived tumors in adults, 88% in children and in 100% in medullary carcinoma in adults, regardless of preoperative vocal fold mobility. Statistically, the only factor correlated with nerve invasion was an aggressive histologic subtype for follicular cell-derived tumors. Knowledge of the pathological subtype may help choose between nerve resection and nerve preservation, if effective adjuvant therapy is available. 

## Figures and Tables

**Figure 1 cancers-14-02961-f001:**
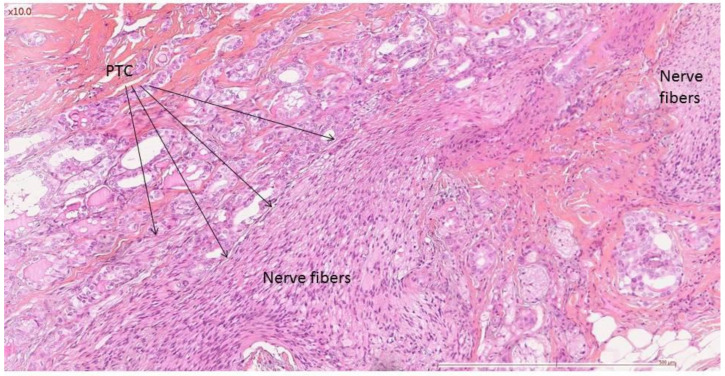
Classic papillary thyroid carcinoma (PTC) in contact with nerve fibers (Schwann cells).

**Figure 2 cancers-14-02961-f002:**
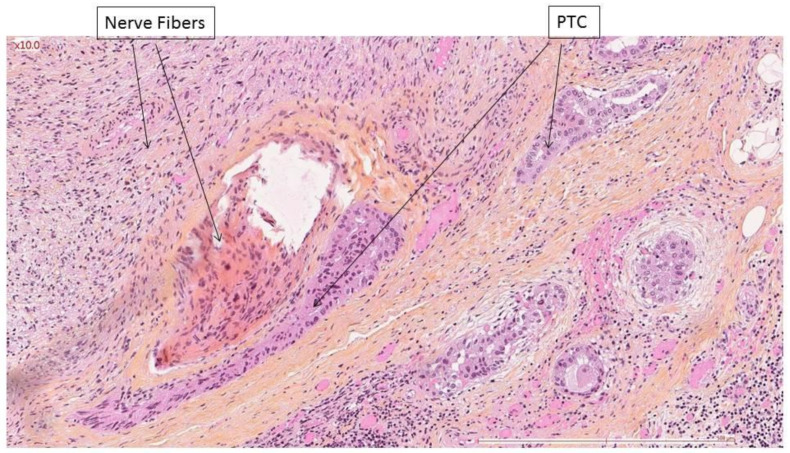
Classic papillary thyroid carcinoma (PTC) within the nerve sheath, in contact with nerve fibers.

**Figure 3 cancers-14-02961-f003:**
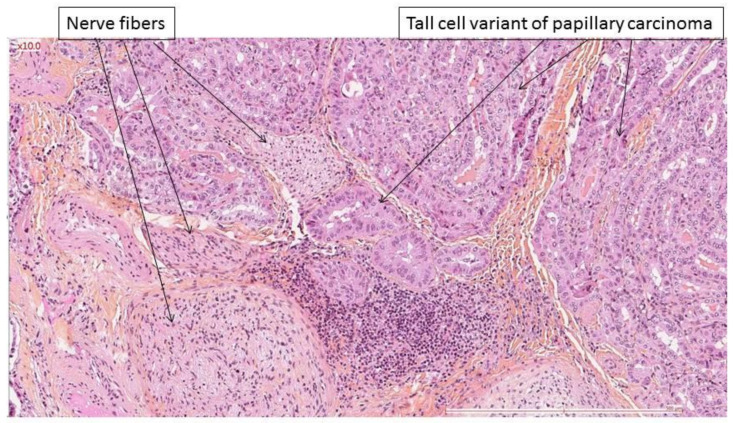
Tall cell variant of papillary thyroid carcinoma invading the nerve, in contact with nerve fibers.

**Table 1 cancers-14-02961-t001:** Relationship between preoperative vocal fold (VF) paralysis and histopathological nerve invasion.

		Preoperative VF Paralysis	Presence of Nerve Invasion	Rate of Nerve Invasion
Follicular cell-derived carcinoma	Adults *n* = 37	No *n* = 23	21/23	30/37 (81%)
		Yes *n* = 14	9/14	
	Children *n* = 8	No *n* = 7	6/7	7/8 (88%)
		Yes *n* = 1	1/1	
Medullary thyroid carcinoma	Adults *n* = 7	No *n* = 6	6/6	7/7 (100%)
		Yes *n* = 1	1/1	

**Table 2 cancers-14-02961-t002:** Factors to consider when choosing nerve preservation versus nerve resection (adapted from References [1,2]).

Patient Factors	Oncologic Factors
Age	RAI+ versus PET+ disease
Young patients have a higher chance of RAI- avid disease Older patients have a higher risk of PET+/RAI- disease	- RAI can be curative for low volume R2 disease
Comorbidities Risk of aspiration pneumonia	Risk of local complications if residual disease (R2) left near the visceral axis
Voice and laryngeal mobility Contralateral vocal fold mobility	Complete resection possible or not
Quality of life choices	Aggressive histology
	Primary or recurrent disease
	Distant metastases
	- indication for systemic therapy
	External beam radiation therapy possible

RAI: Radioactive iodine. PET: 18-fluorodeoxyglucose positron emission tomography.

## Data Availability

The data presented in this study are available on request from the corresponding author.

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
