# Peer review of "Pathological Analysis of Encased Resected Recurrent Nerves in Locally Invasive Thyroid Cancer"

_cancers, 2022, doi:10.3390/cancers14122961_

Round 1
Author Response
Response to reviewer 1
We thank the rewiewer for their time and effort and for their very pertinent critiques that will help us greatly improve the manuscript.
In the revised version that we are submitting, we have added remarks in the text, highlighted in blue, in response to the reviewer:
REVIEWER 1
1
Cancers-1764315
Dahan A, et al. Pathological Analysis of Encased Resected Recurrent Nerves in Locally Invasive Thyroid Cancer.
The authors pathologically examined the presence/absence of microscopic invasion of the recurrent laryngeal nervesresected due to encasement in thyroid cancer.They compared the rate of invasion between various kind of thyroid cancers including common type follicular-cell derived tumors, aggressive subtypes (tall cell, columnar cell, and diffuse sclerosing), pediatriccases and medullary thyroid carcinoma. While, this retrospective
study did not show how many patients could avoid nerve resection by shaving or partial layer resection.In this biased population, the conclusion that aggressive variants and medullary thyroid carcinoma invaded the nerve more often than classic papillary thyroid carcinoma does not seem to be fair. To conclude nerve preserving techniques in the presence of an encased nerve may carry a high risk of leaving significant residual disease, local recurrence rate around the nerve should be shown in cases where the nerve was
preserved by R1/R2 resection.
The reviewer points out a very important problem in our retrospective study: very few patients had R2 resection (leaving macroscopic residual disease), so that we really did not have any groups that would have enabled us to compare outcomes. We have deleted this reference to « clinically significant disease » in the abstract (line 28) and in the conclusion (line 260).
The resected nerves were pathologically invaded in 85% of whole cohort. Thisfact indicates that the decision to resect or preserve the nerve is quite appropriate. The authors only descried the way of decision-making as “multifactorial”. Please specify more concrete methods to choose nerve resection than the generalities in Table 2.
Unfortunately, our study is retrospective and heterogenous, so there is not a simple answer to this question. Each case was discussed preoperatively in our local tumor board and nerve resection (if necessary) was validated. When complete resection (R0/R1) was deemed to be possible by the tumor board (with surgeons attending), nerve resection was validated if necessary to achieve this goal. When the tumor was known to be iodine-refractory (for differentiated thyroid cancer) our tumor board generally also validated nerve resection. Most of these patients were treated before the widespread use of BRAF or RET inhibitors or before neo-adjuvant treatment protocols were employed. Some of these cases today may have been amenable to neo-adjuvant treatment kinase inhibitors, and their nerves could possibly be spared.
We have added these remarks in the discussion (lines 252-262)
How do the authors utilize intraoperative nerve monitoring?
Intraoperative nerve monitoring was used in an attempt to preserve nerves in the absence of preoperative paralysis (nerve shaving, when possible). We also find it helpful when dissecting the contralateral lobe, to predict possible bilateral vocal fold paralysis and be ready to manage it postoperatively.
Surprisingly enough, patients with preoperative recurrent nerve palsy had lower rate of microscopic invasion than who without palsy. Please describe the situations further.
This has been shown in other studies: the nerve can resist invasion due to the fact that the nerve fibers are spread apart by the tumor but not interrupted, and this is particularaly true for relatively slow-growing tumors like thyroid cancer. On the other hand, compression and stretching, and maybe even local vascular compromize by tumor compression by a bulky tumor and/or lymph nodes may induce paralysis without invasion within the nerve sheath (lines 204-209).
In recurrent cases, postoperative adhesion may make it difficult to assess the presence of true invasion intraoperatively. These cases should be described in detail including the status at initial surgery and evaluated separately.
Unfortunately, most of these reoperative patients had been initially treated at oustide institutions and details regarding the initial surgery were missing (initial operating reports were often missing as well). Generally speaking, however, when we have trouble distinguishing between fibrosis and tumor intraoperatively we do rely on frozen section analysis to guide us. In addition, all of these cases had preoperative workup with contrast-enhanced cross-sectional imaging and many with 18 FDG-PET-CT which showed tumor in the area of the nerve (contrast uptake or FDG uptake).
Minor points
1. Study period was between 2005 and 2020 in Abstract, but from 2005 to 2021 in line
48.Please correct.
This has been corrected
2. In this study, is microscopic invasion different from perineural invasion? The definition
of microscopic invasion (line 234-235) should be described in Methodssection.
2
We have made the correction (line 52)
3. Did the two pathologists agree with microscopic invasion in all cases?
Yes, the slides were unequivocal.
4. Statistical methods and software should beshown in Methodssection.
We have made the correction (lines 54-56).
5. Were all follicular-cell derived tumor papillary thyroid carcinoma?
We have added more detail as to the pathology of the tumors (lines 62-64).
6. Continuous variables should be shown mean (average)±standard deviation or median
with range as appropriate (a normal distributionor not).
We have added medians and range (lines 137-140).
Regarding the analysis of risk factors for microscopic invasion, additional table would be
helpful.
There was only one variable that was statistically significant (pathology). We have added the specific variables analyzed for clarity (lines 53-54 and 103-104).

Reviewer 2 Report
Dear Editor,
I read with pleasure the article by A Dahan et al on “pathological analysis of encased resected RLN in locally invasive thyroid cancer”.
The article reports on relatively rare, but very important clinical situations, the invasion of RLN with thyroid cancer. The data is very interesting, well analyzed and well presented. The results are very interesting and, to my knowledge, never presented in such a clear way with so many patients (52 patients aver 15 years, the largest published series of resected RLN because of cancer invasion).
The limitations are clearly mentioned: of course the readers would like to have more informations on the functional follow-up of those patients (nerve reconstruction procedures, medialization procedures, voice outcomes, etc…) but the data is not available. As the authors have a large experience in this area, they could perhaps add a short paragraph on their current practice when they resect the nerve (reconstruction with ansa cervicalis when the distal part is visible, no reconstruction if the distal part is not visible but with early ? late ? medialization, which type ? only if patients are symptomatic ? etc…)
Minor comments:
Lines 52 + 53, the authors refer to Fig 1 with the wording “epineurium”. This wording is not clearly mentioned in the Fig 1 legend. I would use the same wording in the text and in the Fig legend for clarity. In the text, I wonder whether the grammar is correct “ … the presence of tumor cells or stroma IN BEYOND the outer….”Perhaps ok, Im’ not native English speaker !
Line 140: spelling mistake “…and UTLINE the multiple…
Line 176, seplling mistake “…The 2018 INERNATIONAL..Neuromonitoring…”
Author Response
Response to reviewer 2
We thank the rewiewer for their time and effort and for their very pertinent critiques.
In the revised version that we are submitting, we have added remarks in the text, highlighted in yellow, in response to the reviewer:
We actually did contact patients by mail in an attempt to evaluate the long-term functional outcomes, but only 15 patients (adults) responded to the questionnaires. We have added this information, and information about tracheostomy, in the Results section (line 79 was deleted and lines 124-137 have been added) :
From a functional standpoint, four adult patients and two pediatric patients required tracheostomy for concurrent laryngo-tracheal resections, but with only 1 adult patient maintaining a permanent tracheostomy. The two pediatric patients were decannulated after endoscopic laser-assisted medial arytenoidectomy. Intraoperative nerve reconstruction with ansa hypoglossi was performed in only 2 patients due to the distal resection of the nerve (close to the larynx) in most cases, but also due to the fact that in the early patients in our study, this practice was not routine. Sedondary vocal fold medialization was performed in 5 adults—3 autologous fat injections and 2 Isshiki type I thyroplasties [6]—and one child (autologous fat injection). Fifteen adult patients responded to a mailed questionnaire sent an average of 2 years post-operatively for subjective evaluation. The average voice handicap index (VHI 10) [7] for these fifteen patients was 19.7, with a median of 17 (range 4-34). The average score on the European Organization for Research and Treatment of Cancer Quality of Life Questionnaire-Head and Neck 35 (EORTC-QLQ-H&N35) [8] questionnaire was 60, with a median of 52 (range 35-108).
In the discussion we have added information pertaining to the VHI and EORTC scores and to our current practice in terms of nerve reconstruction (lines 231-238):
The VHI and EORTC data were only available for 15 patients who responded to the questionnaires. A VHI 10 score above 11 is considered abnormal. [42] The VHI 10 scores observed (19.7 on average, with a median of 17) are similar to those reported for transoral laser-assisted vocal fold resection for laryngeal cancer. [43] The EORTC scores were comparable to average scores observed in patients treated for head and neck squamous cell carcinoma. [8] Today, we attempt ansa hypoglossi nerve anastomosis in all cases in which a small nerve stump can be preserved at the laryngeal entry point. We do not perform intralaryngeal anastomosis when the tumor involved the distal end of the recurrent nerve, but this technique seems to improve functional results when feasible.[36]
We have added references 6, 7, 8, 42 and 43.
Lines 52 + 53
This has been corrected.
Line 140: spelling mistake “…and UTLINE the multiple…
This has been corrected to read “outline” (line 154 in the revised manuscript)
Line 176, seplling mistake “…The 2018 INERNATIONAL..Neuromonitoring…”
This has been corrected (line 190 in the revised manuscript)

Round 2
Reviewer 1 Report
I think the authors made appropriate revision of the manuscript and the paper would fulfill the standard of the journal.